# A Study on the Perceptions of Korean Older Adult Patients and Caregivers about Polypharmacy and Deprescribing

**DOI:** 10.3390/ijerph191811446

**Published:** 2022-09-11

**Authors:** Han-Gyul Lee, Seungwon Kwon, Bo-Hyoung Jang, Jin Pyeong Jeon, Ye-Seul Lee, Woo-Sang Jung, Sang-Kwan Moon, Ki-Ho Cho

**Affiliations:** 1Kyung Hee University Medical Center, Department of Cardiology and Neurology, College of Korean Medicine, Kyung Hee University, Seoul 02447, Korea; 2Department of Preventive Medicine, College of Korean Medicine, Kyung Hee University, Seoul 02447, Korea; 3Department of Neurosurgery, Hallym University College of Medicine, Chuncheon 24253, Korea; 4Jaseng Spine and Joint Research Institute, Jaseng Medical Foundation, Seoul 06110, Korea

**Keywords:** polypharmacy, deprescribing, older adults, survey, questionnaire

## Abstract

Polypharmacy is continuously increasing among older adults. The resultant potentially inappropriate medications (PIMs) can be harmful to patient health. Deprescribing refers to stopping or reducing PIMs. In this study, the current status of polypharmacy and willingness of older adults to deprescribe were investigated among patients and caregivers who are not associated with one another. The survey used the Korean translated version of the revised Patients’ Attitude Towards Deprescribing (rPATD) Scale. Data were collected through an online survey of 500 participants (250 patients and caregivers each) in this study. The following results were found for patients and caregivers, respectively: 74.8% and 63.6% felt their number of medications was high, 64.4% and 55.6% desired to reduce their medications, 70.4% and 60.8% were concerned about medication discontinuation, 63.2% and 61.2% had a good understanding of their medications, 77.6% and 76.4% were willing to be well informed, and 79.6% and 72% wanted to reduce the number of medications if medically feasible. Patients and caregivers commonly agreed to the burden of the number of medications they were taking, and were willing to reduce the number of medications if the doctor said it was possible. Doctors should consider this information during the deprescribing process, and promote deprescription while involving patients and caregivers in the decision-making process.

## 1. Introduction

Polypharmacy is generally defined as the use of five or more medications [1,2]. Due to the increase in chronic diseases caused by aging, polypharmacy continues to increase worldwide, indicating that approximately 50% of people aged 65 years or older are taking five or more medications [3,4,5]. In Korea, the prevalence of polypharmacy among those aged ≥65 years was 44.1–86.4% [2,6,7]. Polypharmacy can be appropriate and necessary in some patients, but the polypharmacy associated with potentially inappropriate medications (PIMs) may result in the medication’s inherent “potential benefit” being of “potential harm” instead, due to interaction among medications or the resulting increase in adverse effects [8,9]. Polypharmacy associated with PIMs may increase the risk of adverse drug reactions, hospitalization, reduced quality of life and mortality [10,11,12,13].

Deprescribing is defined as the process of withdrawal or dose reduction of PIMs supervised by a healthcare professional [14]. Several studies have shown that older patients with polypharmacy want to deprescribe if medically feasible [15,16,17,18,19,20]. However, deprescribing may not occur as often as it should in practice, because of barriers between prescribers and patients [21]. Patient-related barriers include beliefs about the necessity of medications, bad experiences with stopping, influence of doctors/family/friends, and fear of withdrawal effects [22]. Therefore, there is a need to investigate older patients’ attitudes, beliefs, and experiences regarding polypharmacy and their willingness to deprescribe. In addition, most older patients have a relatively poor understanding of medications because of their low health literacy [23,24]. Thus, for more accurate information, the same investigation for caregivers such as the families of older patients is also needed. 

The Patients’ Attitudes Towards Deprescribing (PATD) Questionnaire was developed and validated to assess people’s attitudes, beliefs, and experiences regarding the number of medications they are taking and their feelings about stopping or reducing medications [25]. More than 60% of the patients who completed this survey as part of a previous study felt that they were taking a large number of medications, and 92% stated that they would be willing to stop one or more of their medications if possible [20]. However, the PATD has a limitation in that there is no scoring system or version for caregivers or family members. The revised Patients’ Attitudes Towards Deprescribing (rPATD) questionnaire has expanded in scope from the original PATD, and includes both a scoring utility and caregivers’ version [26]. In a previous study examining the perception of patients and caregivers, 87.6% of patients and 74.8% of caregivers desired deprescribing [27]. However, to the best of our knowledge, no study has investigated the deprescribing attitudes of patients and caregivers in Korea. 

This study aimed to investigate the current status of polypharmacy, and the willingness to deprescribe among patients and caregivers in the general older adult population currently undergoing polypharmacy in Korea. The main aim is to provide suggestions for improving polypharmacy and develop related treatment technologies for older adults in the future. 

## 2. Materials and Methods

### 2.1. Participants Recruitment

This study was conducted with patients using polypharmacy, or caregivers of patients using polypharmacy. The patient inclusion criteria were as follows: (1) 65 years of age or above, and (2) six or more medications currently being taken [2]. The caregiver inclusion criteria were as follows: (1) the care recipient is 65 years of age or older; (2) the care recipient is currently taking six or more medications [2]. There was no restriction on types and administration methods of medications for inclusion in the survey. Caregivers were limited to family members, and did not have an employment relationship. Participants who understood the purpose of this study and voluntarily consented to participate were included. Cases in which the survey was impossible due to communication difficulties were excluded. Participants constituted the self-owned survey panel of Medi Research, a survey agency, and if they agreed to participate, the survey was conducted online. The self-owned survey panel was capable of establishing opinions corresponding to the average age, sex, and region of Korea. The study protocol was approved by the Institutional Review Board of Kyung Hee University (KHSIRB-21-455(RA), approved on 21 October 2021) and was conducted according to the guidelines of the Declaration of Helsinki.

In this study, 500 participants (250 older patients and 250 care givers) were recruited by convenience in consideration of the study size and feasibility. When participating in the survey, a participation compensation fee was paid.

### 2.2. Survey

The survey examined attitudes toward deprescribing used the rPATD developed for older adult (≥65 years old) patients and caregivers of older adult patients taking at least one regular medication [26]. This questionnaire was completed in English as it was developed in Australia; therefore, to use it for older participants in Korea, a translation process with a questionnaire validation was required. The rPATD questionnaire has previously been translated and cross-culturally adapted into other languages [28,29,30]; three versions in French [31], Arabic [32], and Danish [33] have been thoroughly validated. However, as the Korean version of the rPATD was unavailable, we used a version that was translated into Korean by two researchers fluent in both Korean and English (SK and WSJ). Participants’ characteristics and current polypharmacy were also investigated, along with the rPATD. 

### 2.3. Data Collection and Recording

The questionnaire, translated by the researchers, was delivered to Medi Research and developed for an online survey through web programming. The online questionnaire was developed by transmitting the link through the contact information and e-mail of the panel. When the panel agreed to participate, consent within the link was obtained, and the participants responded directly to the online questionnaire. After completing the questionnaire, the data were transmitted and collected by Medi Research, and raw data were delivered to the research team after data processing. 

### 2.4. Data Analysis

Descriptive statistics were used to analyze the results. This analysis was performed using Microsoft Excel, version 15.26. All data are expressed as a number (%).

## 3. Results

### 3.1. Participant Characteristics

A total of 500 patients (250 patients and caregivers each) participated in the survey. Most patients were between 65 and 69 years of age, with the exception of two patients aged between 70 and 79 years. The caregivers’ ages were evenly distributed from 20s to 60s, with the 30s (74 with 29.6%) and 40s (90 with 36%) being the most common. Of the patients, 144 (57.6%) were male; among the caregivers, 123 (49.2%) were male. The participant demographics are presented in Table 1.

### 3.2. Recognition of the Current State of Polypharmacy

Table 2 shows the current polypharmacy status of the patients and caregivers. The number of regular medications, including Western medications prescribed by Medical Doctors (MD), herbal medications prescribed by a Doctor of Korean Medicine (DKM), health functional foods approved by the Korea Ministry of Food and Drug Safety (KFDA), and health foods which are not approved by KFDA, were the most common in both groups at six (patients: 135, 54%, caregivers; 217 86.8%). Hypertension was the most common disease or symptom in both groups (patients: 175 [70%], caregivers: 172 [68.8%]). In the patient group, hyperlipidemia, digestive disorders, pain symptoms, and diabetes mellitus were the next most common, and in the caregiver group, diabetes mellitus, hyperlipidemia, digestive disorders, and pain symptoms were the next most common. 

### 3.3. Patients’ Perception of Deprescribing

Table 3 and Figure 1 show patients’ rPATD responses. Most of the patients felt that the number of medications was too much (187 [74.8%] in response to burden item 3: “I feel that I am taking a large number of medicines”). More than half of the patients agreed with the financial burden of taking medications (136 [54.4%] in response to burden item 1: “I spend a lot of money on medicines”). Further, many patients were hoping their doctor would reduce the number of medications (161 [64.4%] in response to appropriateness item 3: “I would like my doctor to reduce the dose of one or more of my medicines”). Meanwhile, some patients were hesitant to adjust the number of medications (176 [70.4%] in response to concerns about stopping item 1: ”I would be reluctant to stop a medicine that I had been taking for a long time”); fear of loss of benefits was presumed to be the reason for this (168 [67.2%] in response to concerns about stopping item 2: ”If one of my medications was stopped, I would be worried about missing out on future benefits”). Patients were well aware of their own medications (158 [63.2%] in response to involvement item 1: ”I have a good understanding of the reasons I was prescribed each of my medicines”) and were also willing to be well informed (194 [77.6%] in response to involvement item 3: “I like to know as much as possible about my medicines”). Overall, patients tended to reduce the number of medications if this had been confirmed by professional medical personnel (199 [79.6%] in global item 1: “If my doctor said it was possible, I would be willing to stop one or more of my regular medicines”). 

### 3.4. Caregivers’ Perception of Deprescribing

Table 4 and Figure 1 show caregivers’ rPATD responses. Many caregivers felt that the number of medications was too much (159 [63.6%] in burden item 3: ”I feel that the person I care for is taking a large number of medicines”). More than half of caregivers agreed with the financial burden of patients taking medications (133 [53.2%] in response to burden item 1: ”My care recipient’s medicines are quite expensive”). Further, many caregivers were hoping the doctor would reduce the number of medications (139 [55.6%] in response to appropriateness item 3: ”I would like the doctor to reduce the dose of one or more of my care recipient’s medicines”). However, caregivers tended to be hesitant to adjust the number of medications (152 [60.8%] in concerns about stopping item 1: ”I would be reluctant to stop one of my care recipient’s medicines that they had been taking for a long time”), and fear of loss of benefits was presumed to be the reason (166 [66.4%] in response to concerns about stopping item 2: ”If one of my care recipient’s medicines was stopped, I would be worried about missing out on future benefits”). Caregivers were well aware of their care recipients’ medications (153 [61.2%] in involvement item 1: ”I have a good understanding of the reasons why my care recipient was prescribed each of their medicines”) and were also willing to be well informed (191 [76.4%] in response to involvement item 3: ”I like to know as much as possible about my care recipient’s medicines”). Overall, caregivers wished to reduce the number of medications if confirmed by professional medical personnel (180 [72%] in response to global item 1: ”If their doctor said it was possible, I would be willing to stop one or more of my care recipient’s medicines”). These results are similar to those of the patient group.

## 4. Discussion

To the best of our knowledge, this is the first study on attitudes towards polypharmacy and deprescribing among older patients and caregivers in Korea. Regarding burden items, more than half of the patients responded that they felt burdened on all items, and in particular, the feeling that they were taking too many medications was the most prominent. The caregivers’ responses showed similar results. Previous studies using PATD also reported that older patients felt that they were taking many medications at a similar rate [15,18]. The proportion of polypharmacy in older patients is high at approximately 50% [3,4,5], and the increase in older adult patients with various complex chronic diseases according to life extension is cited as the main reason for the current increase in polypharmacy worldwide [5]. In addition, treatment patterns based on single-disease guidelines are another major cause of increased polypharmacy [34]. As the existing guidelines were not developed for older patients with multiple chronic diseases, it is presumed that under the current medical guidelines, older patients are likely to be exposed to polypharmacy, which leads to an increase in the burden of medications. 

On studying the decline in economic income and increase in medical expenditure among older adults, we see that approximately 20% of income is spent on medical expenses [35,36]; prescription medication expenditures account for a large proportion [37]. In this study, most patients and caregivers responded that they experience financial burden, which is thought to have a significant impact on the burden of polypharmacy. Korea’s medical insurance system is the National Health Insurance Service (NHIS), and medications for underlying diseases such as hypertension, diabetes, and hyperlipidemia, which are frequent medications for older adults, are mostly covered by this insurance. However, the NHIS has a self-pay ratio that can be relatively burdensome for older adults. In addition, there are blind spots in the NHIS, with 12% of respondents in the present study saying that they do not have NHIS. Unlike the NHIS, the UK’s National Health Service (NHS) pays full medical expenses, including medication expenditure, for those aged 60 years or older. Therefore, older adults in the UK think that the NHS is spending a lot of money on their medications; however, this has not led to a burden or desire to stop medications [38]. Although private insurance can supplement the NHIS, the proportion of older adults availing private insurance is lower than in other age groups, and socioeconomic factors have been found to play a role in this [39]. Among the patients in this study, the private insurance subscription rate was less than 40%. 

The desire for doctors to reduce the number of medications was common among the respondents, but there was a tendency to be reluctant to stop taking the medication for a long time. There was also a fear of future benefits that might be lost due to discontinuation of the medication. The main reasons for the barrier to patient deprescribing are “appropriateness” and “fear” for medication cessation [22], which can tear down the barrier to accurate understanding of medication currently being taken by patients and why it should be discontinued. For the involvement item, both patients and caregivers had a high level of understanding and willingness to know the medications they were currently taking, and tended to actively participate in the prescription process (Figure 1). Considering that studies with a higher acceptance rate of deprescribing had more involvement of the participant’s general practitioner than the other studies [40], doctors may need to accept patients’ willingness to participate in deprescribing. However, patients tend to comply with doctors’ decisions despite concerns [41] about deprescribing or being reluctant [42] due to the possibility of adverse outcomes, even if they participate in the decision-making process [38]. 

Overall, approximately 80% of the patients in this study were willing to deprescribe if medically feasible. This result was consistent with a previous study that investigated hospitalized patients using rPATD [38], and was similar to the willingness ratio (83–94%) for deprescribing that was investigated using PATD for patients alone [15,16,17,18,19,20]. However, considering that taking antiepileptics, anti-Parkinson agents, and analgesics showed a low response to deprescribing [43], while antihypertensive agents, lipid-modifying agents, antithrombotic agents, and osteoporosis medications showed relatively high acceptance [44], it seems that different types of medications may show different acceptance levels. In the current study, the high rate of antihypertensive medication use (which showed the highest levels of lipid-modifying agents and osteoporosis treatments) may have affected the positive response to deprescribing. 

As older patients may have low health literacy [23,24] or cognitive decline, caregivers often become proxy decision-makers for medication withdrawal [45]. Therefore, it is necessary to understand caregivers’ views on deprescribing. The perception of deprescribing was positive in both groups, but slightly lower at 72% among caregivers, compared to approximately 80% among patients. This is consistent with a previous study that showed that caregivers prefer a more passive role in medical decision-making [27]. For the burden item, the burden felt by the patients was higher overall. As medication-related burden plays a central role in affecting the patient’s health and daily lives, their beliefs, and behaviors about medications [46], it can be inferred that the burden directly felt by the patients was higher. Caregivers’ lack of understanding of patients’ treatments may lead to disagreement with the patient about the appropriateness of medications [47]. The caregivers in the current study were aware of the treatment of their care recipients and had a high willingness to know about the medications, which could be attributed to the lack of disagreement between the two groups regarding appropriateness. 

The polypharmacy solution proposed in this study is as follows. Disease-oriented prescriptions based on the current single disease guidelines are highly likely to cause polypharmacy in older adult patients with multiple chronic diseases [34]. Therefore, from a patient-oriented perspective, there is a need for a clear guideline for deprescription and medical staff to periodically adjust medications stepwise. The United States and Europe have produced the Beers criteria [48] and STOPP/START criteria [49], respectively, as guidelines for PIM’s deprescription. However, it significantly differs from the medications distributed in Korea; therefore, there is an urgent need to establish a domestic equivalent. In particular, single disease guidelines conflict with each other, and a multidisciplinary approach will be helpful when establishing domestic evidence. Monitoring of duplicate prescriptions is also necessary. There is a need for a concrete plan to increase the utilization of the existing drug utilization review, which is currently used to check whether there are duplicate prescriptions. As cardiovascular medications, such as statins, antidepressants, and proton pump inhibitors have been reported as representative medications with frequent duplicate prescriptions [50,51], monitoring these medications and reducing duplicate prescriptions will be helpful. Finally, as the participation of doctors, patients, and broadly everyone involved in the medical process can lead to changes in deprescription [52], policies to encourage wider participation are needed. The results of this study may be significant as a first step in these discussions. 

This study has a number of limitations. First, the Korean version of the rPATD used in this study was translated by multiple researchers fluent in both Korean and English, but did not go through the process of cross-cultural adaptation and validation. Therefore, development and validation of the Korean version of the rPATD are required. Second, all of the surveys were conducted non-face-to-face online. There are methodological limitations in online surveys as they cannot explain the distributed population, and respondents with biases can select themselves as samples [53]. A direct face-to-face study using the developed Korean version of the rPATD is required in the future. Lastly, there is a limitation in that the results of the present study were conducted on relatively young older adults (all but two were aged 65-69 years old). The present study was conducted through an online survey, so we assumed that it was relatively difficult for those in their 70s or older to participate. In addition, due to the nature of the online survey, if there was a decrease in consciousness or severe cognitive decline, it was judged that there was a communication difficulty, and participation was limited. As a result, there is a limitation that the perception of the older adults in a severe condition was not reflected in the results of the present study. Further studies on discrepancies between medical staff and patient caregivers are also required through surveys of the medical staff. 

## 5. Conclusions

Patients and caregivers commonly agreed to the burden of the number of medications they were taking, and were willing to reduce the number of medications if the doctor said it was possible. The financial burden on older adults also contributed significantly to the burden of taking medications. In addition, there was psychological resistance to the decision to discontinue medication, and a high willingness to participate and an understanding of treatment could overcome this barrier. Therefore, doctors should enable patients and caregivers to actively participate in the deprescribing and decision-making processes, and discuss in detail the possibility and implementation of deprescribing with them. 

## Figures and Tables

**Figure 1 ijerph-19-11446-f001:**
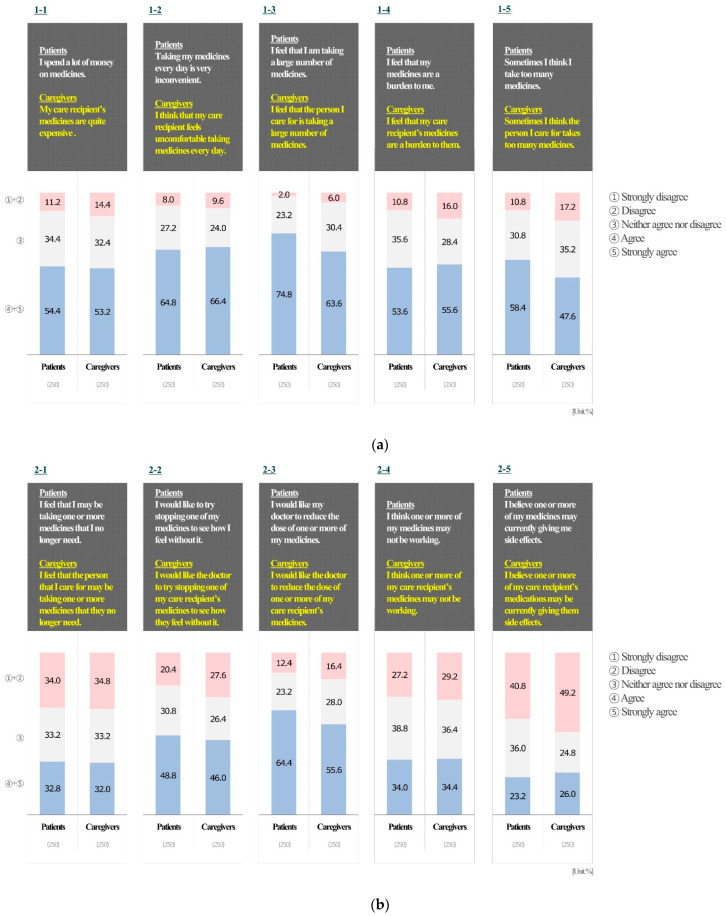
Overview of patients’ and caregivers’ responses to the rPATD questionnaire. (**a**) burden factor, (**b**) appropriateness factor, (**c**) concerns about stopping factor, (**d**) involvement factor, and (**e**) global factor.

**Table 1 ijerph-19-11446-t001:** Participant characteristics.

Characteristics	Patients (*n* = 250)	Caregivers (*n* = 250)
**Age (years)**		
65–69	248 (99.2)	
70–79	2 (0.8)	
20–29		8 (3.2)
30–39		74 (29.6)
40–49		90 (36)
50–59		29 (11.6)
60–69		49 (19.6)
**Sex (male)**	144 (57.6)	123 (49.2)
**Highest education completed**		
High school graduation or below	53 (21.2)	46 (18.4)
Bachelor’s	176 (70.4)	179 (71.6)
Postgraduate	21 (8.4)	25 (10)
**Income per month (won)**		
≤3 million	69 (27.6)	63 (25.2)
3–5 million	84 (33.6)	89 (35.6)
≥5 million	97 (38.8)	98 (39.2)
**Place of residence**		
Home alone	34 (13.6)	39 (15.6)
Home with spouse	156 (62.4)	123 (49.2)
Home with children	51 (20.4)	56 (22.4)
Home with other family members	9 (3.6)	22 (8.8)
Facilities	0 (0)	10 (2.5)
**Type of insurance**		
National Health Insurance Service	220 (88)	
Medical aid	30 (12)	
Private health insurance	98 (39.2)	

Note: Data are expressed as number (%).

**Table 2 ijerph-19-11446-t002:** Current status of medication use.

	Patients (*n* = 250)	Caregivers (*n* = 250)
**Number of regular medications** ** ^†^ **		
6	135 (54)	217 (86.8)
7	48 (19.2)	21 (8.4)
8	27 (10.8)	4 (1.6)
9	12 (4.8)	2 (0.8)
10 or more	28 (11.2)	6 (2.4)
**Taking western medications**	250 (100)	245 (98)
**Taking herbal medications**	71 (28.4)	88 (35.2)
**Consuming health functional foods**	212 (84.8)	216 (86.4)
**Consuming health foods**	203 (81.2)	206 (82.4)
**Causes of using medications**		
Hypertension	175 (70)	172 (68.8)
Diabetes mellitus	95 (38)	108 (43.2)
Hyperlipidemia	131 (52.4)	89 (35.6)
Digestive disorders	107 (42.8)	76 (30.4)
Pain symptoms	100 (40)	63 (25.2)
Osteoporosis	69 (27.6)	56 (22.4)
Sleep disorders	50 (20)	53 (21.2)
Heart diseases	62 (24.8)	39 (15.6)
Stroke	47 (18.8)	29 (11.6)
Degenerative brain diseases	49 (19.6)	29 (11.6)
Respiratory diseases	48 (19.2)	29 (11.6)
Mental symptoms	34 (13.6)	20 (8)
Other diseases	6 (2.4)	11 (4.4)

Data are expressed as a number (%). † Medications including Western medicines, herbal medicines, health functional foods, and health foods.

**Table 3 ijerph-19-11446-t003:** Patients’ responses to the rPATD questionnaire about awareness of current medication use status.

Item	Strongly Disagree	Disagree	Neither Agree Nor Disagree	Agree	Strongly Agree
**Burden factor**					
1. I spend a lot of money on medicines.	3 (1.2)	25 (10)	86 (34.4)	**109 (43.6)**	27 (10.8)
2. Taking my medicines every day is very inconvenient.	-	20 (8)	68 (27.2)	**121 (48.4)**	41(16.4)
3. I feel that I am taking a large number of medicines.	-	5 (2)	58 (23.2)	**135 (54)**	52 (20.8)
4. I feel that my medicines are a burden to me.	1 (0.4)	26 (10.4)	89 (35.6)	**106 (42.4)**	28 (11.2)
5. Sometimes I think I take too many medicines.	1 (0.4)	26 (10.4)	77 (30.8)	**123 (49.2)**	23 (9.2)
**Appropriateness factor**					
1.I feel that I may be taking one or more medicines that I no longer need.	6 (2.4)	79 (31.6)	**83 (33.2)**	76 (30.4)	6 (2.4)
2. I would like to try stopping one of my medicines to see how I feel without it.	3 (1.2)	48 (19.2)	77 (30.8)	**99 (39.6)**	23 (9.2)
3. I would like my doctor to reduce the dose of one or more of my medicines.	3 (1.2)	28 (11.2)	58 (23.2)	**120 (48)**	41 (16.4)
4. I think one or more of my medicines may not be working.	3 (1.2)	65 (26)	**97 (38.8)**	71 (28.4)	14 (5.6)
5. I believe one or more of my medicines may currently be giving me side effects.	15 (6)	87 (34.8)	**90 (36)**	55 (22)	3 (1.2)
**Concerns about stopping factor**					
1. I would be reluctant to stop a medicine that I had been taking for a long time.	6 (2.4)	11 (4.4)	57 (22.8)	**143 (57.2)**	33 (13.2)
2. If one of my medications was stopped, I would be worried about missing out on future benefits.	3 (1.2)	17(6.8)	62 (24.8)	**139 (55.6)**	29 (11.6)
3. I get stressed whenever changes are made to my medicines.	4 (1.6)	34 (13.6)	**97 (38.8)**	94 (37.6)	21 (8.4)
4. If my doctor recommended stopping a medicine, I would feel that he/she was giving up on me.	10 (4)	79 (31.6)	**92 (36.8)**	57 (22.8)	12 (4.8)
5. I have had a bad experience when stopping a medicine before.	15 (6)	79 (31.6)	**81 (32.4)**	63 (25.2)	12 (4.8)
**Involvement factor**					
1. I have a good understanding of the reasons I was prescribed each of my medicines.	1 (0.4)	10 (4)	88 (35.2)	**136 (54.4)**	22 (8.8)
2. I know exactly what medicines I am currently taking, and/or I keep an up-to-date list of my medications.	-	19 (7.6)	**106 (42.4)**	**106 (42.4)**	19 (7.6)
3. I like to know as much as possible about my medicines.	1 (0.4)	6 (2.4)	49 (19.6)	**154 (61.6)**	40 (16)
4. I like to be involved in making decisions about my medicines with my doctors.	1 (0.4)	10 (4)	99 (39.6)	**111 (44.4)**	29 (11.6)
5. I always ask my doctor, pharmacist, or other healthcare professional if there is something I don’t understand about my medicines.	1 (0.4)	18 (7.2)	64 (25.6)	**139 (55.6)**	28 (11.2)
**Global factor**					
1. If my doctor said it was possible, I would be willing to stop one or more of my regular medicines.	1 (0.4)	2 (0.8)	48 (19.2)	**160 (64)**	39 (15.6)
2. Overall, I am satisfied with my current medicines.	-	13 (5.2)	**123 (49.6)**	104 (41.6)	9 (3.6)

Note. Data are expressed as number (%). The response with the highest frequency for each item is indicated in bold.

**Table 4 ijerph-19-11446-t004:** Caregivers’ responses to the rPATD questionnaire about awareness of current medications use status.

Item	Strongly Disagree	Disagree	Neither Agree Nor Disagree	Agree	Strongly Agree
**Burden factor**					
1. My care recipient’s medicines are quite expensive.	4 (1.6)	32 (12.8)	81 (32.4)	**115 (46)**	18 (7.2)
2. I think that my care recipient feels uncomfortable taking medicines every day.	3 (1.2)	21 (8.4)	60 (24)	**134 (53.6)**	32 (12.8)
3. I feel that the person I care for is taking a large number of medicines.	-	15 (6)	76 (30.4)	**129 (51.6)**	30 (12)
4. I feel that my care recipient’s medicines are a burden to them.	3 (1.2)	37 (14.8)	71 (28.4)	**123 (49.2)**	16 (6.4)
5. Sometimes I think the person I care for takes too many medicines.	5 (2)	38 (15.2)	88 (35.2)	**94 (37.6)**	25 (10)
**Appropriateness factor**					
1. I feel that the person that I care for may be taking one or more medicines that they no longer need.	7 (2.8)	80 (32)	**83 (33.2)**	73 (29.2)	7 (2.8)
2. I would like the doctor to try stopping one of my care recipient’s medicines to see how they feel without it.	7 (2.8)	62 (24.8)	66 (26.4)	**100 (40)**	15 (6)
3. I would like the doctor to reduce the dose of one or more of my care recipient’s medicines.	8 (3.2)	33 (13.2)	70 (28)	**117 (46.8)**	22 (8.8)
4. I think one or more of my care recipient’s medicines may not be working.	10 (4)	63 (25.2)	**91 (36.4)**	68 (27.2)	18 (7.2)
5. I believe one or more of my care recipient’s medications may be currently giving them side effects.	20 (8)	**103 (41.2)**	62 (24.8)	55 (22)	10 (4)
**Concerns about stopping factor**					
1. I would be reluctant to stop one of my care recipient’s medicines that they had been taking for a long time.	3 (1.2)	23 (9.2)	72 (28.8)	**130 (52)**	22 (8.8)
2. If one of my care recipient’s medicines was stopped, I would be worried about missing out on future benefits.	3 (1.2)	21 (8.4)	60 (24)	**142 (56.8)**	24 (9.6)
3. I get stressed whenever changes are made to my care recipient’s medicines.	7 (2.8)	49 (19.6)	83 (33.2)	**98 (39.2)**	13 (5.2)
4. I feel that if I agreed to stop one of my care recipient’s medicines then this is like giving up on them.	23 (9.2)	**78 (31.2)**	77 (30.8)	58 (23.2)	14 (5.6)
5. The person that I care for has had a bad experience when stopping a medicine before.	16 (6.4)	**80 (32)**	**80 (32)**	62 (24.8)	12 (4.8)
**Involvement factor**					
1. I have a good understanding of the reasons why my care recipient was prescribed each of their medicines.	2 (0.8)	12 (4.8)	83 (33.2)	**134 (53.6)**	19 (7.6)
2. I know exactly what medicines the person that I care for is currently taking, and/or I keep an up-to-date list of their medicines.	1 (0.4)	22 (8.8)	94 (37.6)	**115 (46)**	18 (7.2)
3. I like to know as much as possible about my care recipient’s medicines.	1 (0.4)	4 (1.6)	54 (21.6)	**161 (64.4)**	30 (12)
4. I like to be involved in making decisions about my care recipient’s medicines.	2 (0.8)	17 (6.8)	96 (38.4)	**115 (46)**	20 (8)
5. I always ask the doctor, pharmacist, or other healthcare professional if there is something I don’t understand about my care recipient’s medicines.	1 (0.4)	20 (8)	81 (32.4)	**125 (50)**	23 (9.2)
**Global factor**					
1. If their doctor said it was possible, I would be willing to stop one or more of my care recipient’s medicines.	-	6 (2.4)	64 (25.6)	**155 (62)**	25 (10)
2. Overall, I am satisfied with my care recipient’s current medicines.	1 (0.4)	18 (7.2)	**113 (45.2)**	108 (43.2)	10 (4)

Note. Data are expressed as number (%). The response with the highest frequency for each item is indicated in bold.

## Data Availability

The data presented in this study are available on request from the corresponding author. The data are not publicly available due to protection of personal information.

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
