# Peer review of "A Study on the Perceptions of Korean Older Adult Patients and Caregivers about Polypharmacy and Deprescribing"

_ijerph, 2022, doi:10.3390/ijerph191811446_

Round 1
Reviewer 1 Report
This is a well-written article of health importance. If some methodological descriptions are supplemented, it will be a better manuscript.
#1 Inclusion criteria
Why 6 or more medications? Is there a reference about definition about PIMs pr polypharmacy?
#2 Participant recruitment
How was the participants recruitment conducted? Online advertisement? Please describe the process briefly.
#3 Health insurance status
Most Korean are belong to the National Health Insurance service or Medical aid.
Therefore, the percentage “88%”, require more explanation. What does remained 12% mean?
Are they foreigners? Or Medical aid?
I don't understand how 12% of people don't have health insurance, so it seems necessary to explain that part.
#4 Definition of herbal medicine
What does exactly herbal medicine mean? Only included herbal decoction or extract prescribed by Korean Medicine doctor? or Including health functional food? Or including botanical/ethnopharmacological medicinal plant?
I think more detailed definition about herbal medicine is required in Table 2.
#5 The difference in perception between caregivers and patients is not visible at a glance. A table or figure to show this seems to be necessary.
#6 It is necessary to describe how the results of overseas surveys differ from those in Korea in detail, in the results or discussion section.
Reviewer 2 Report
Please see the attached file.
